# Fermi-liquid corrections to the intrinsic anomalous Hall conductivity of topological metals

Ivan Pasqua[1*], Michele Fabrizio[1]

**1** International School for Advanced Studies (SISSA), Via Bonomea 265, I-34136 Trieste, Italy

\* ipasqua@sissa.it

August 29, 2024

## Abstract

We show that topological metals lacking time-reversal symmetry have an intrinsic non-quantised component of the anomalous Hall conductivity which is contributed not only by the Berry phase of quasiparticles on the Fermi surface, but also by Fermi-liquid corrections due to the residual interactions among quasiparticles, the Landau $f$-parameters. These corrections pair up with those that modify the optical mass with respect to the quasiparticle effective one, or the charge compressibility with respect to the quasiparticle density of states. Our result supports recent claims that the correct expressions for topological observables include vertex corrections besides the topological invariants built just upon the Green's functions. Furthermore, it demonstrates that such corrections are already accounted for by Landau's Fermi liquid theory of topological metals, and have important implications when those metals are on the verge of a doping-driven Mott transition, as we discuss.

# 1 Introduction

In 2004, Haldane showed[1], elaborating on earlier works [2, 3, 4, 5], that in metals where time-reversal symmetry is broken Landau's Fermi liquid theory [6] must be supplemented by a new, topological ingredient: the quasiparticle Berry phase. Specifically, he demonstrated [1] that the expression for the non-quantised component of the intrinsic anomalous Hall conductivity, e.g., in ferromagnetic metals [7], is genuinely a property of the quasiparticle Fermi surface, as later confirmed in *ab initio* calculations [8], and in agreement with the spirit of Landau's Fermi liquid theory [6].

However, Haldane's work leaves open an important issue that we know discuss. Landau's energy functional for quasiparticles [6] includes a quasiparticle energy, which we hereafter denote as the Hamiltonian matrix $\hat{H}_*(\mathbf{k})$ having in mind a multiband Fermi liquid, and a residual interaction among quasiparticles, defined in terms of the so-called Landau $f$-parameters that we write as a tensor $\hat{f}(\mathbf{k}, \mathbf{k}')$ in band space. It is tempting to assume that the quasiparticle topology is encoded just in the quasiparticle Hamiltonian $\hat{H}_*(\mathbf{k})$ and its Bloch eigenstates $|\psi_n(\mathbf{k})\rangle$, a relation that was already put forth by Haldane [1] and later adopted [9]. Nonetheless, there is evidence that such reasonable choice may not be correct [10, 11], namely that the Berry curvature calculated through the quasiparticle Bloch states $|\psi_n(\mathbf{k})\rangle$ not necessarily yields the anomalous Hall conductivity. It is legitimate to wonder whether this evidence is still compatible with Landau's Fermi liquid theory, namely, if the latter does account for Fermi liquid corrections to the anomalous Hall conductivity. This is precisely the aim of the present study.

The structure of the paper is the following. In Sec. 2 we introduce the multi-band formalism of Landau's Fermi-liquid theory and we derive the Hall conductivity tensor, including the Fermi liquid corrections due to the residual interactions between the quasiparticles. In Sec. 3 we specialize our discussion to a model topological metal with broken time-reversal symmetry, where we can explicitly verify that the corrections to the anomalous Hall conductivity stem solely from the quasiparticle Fermi surface. In Sec. 4 we explore the implications of these Fermi liquid corrections in the computation of the topological observables, specifically in strongly correlated topological metals on the verge of a doping-driven Mott transition.

# 2 Hall conductivity within Landau's Fermi liquid theory

We assume a periodic system of interacting electrons, thus in absence of impurities and their extrinsic contributions to the anomalous Hall effect [7], at sufficiently low temperature to safely discard any quasiparticle decay rate, and, as mentioned, we consider the general case of many bands crossing the chemical potential that can be described within Landau's Fermi-liquid theory. However, we shall deal with this theory following an early observation by Leggett [12]. Indeed, one may realise [12, 13] that Landau's Fermi-liquid theory basically shows [14] that the low-frequency, long-wavelength and low-temperature response functions of the physical electrons correspond to those of a system of interacting quasiparticles treated by the Hartree-Fock (HF) plus the random phase (RPA) approximations, apart from important caveats that we discuss in Appendix A.

Therefore, let us consider quasiparticles described by an interacting Hamiltonian, hereafter

setting $\hbar = 1$,

$$H_{\rm qp} = H_0 + H_{\rm int} \, , \tag{1}$$

where

$$H_0 = \sum_{\mathbf{k}} \sum_{\alpha\beta} c^\dagger_{\alpha\mathbf{k}} \, H_{0\,\alpha\beta}(\mathbf{k}) \, c_{\beta\mathbf{k}} \, ,$$

is a one-body term, with $c_{\alpha\mathbf{k}}$ the annihilation operators of the quasiparticles, and $H_{\rm int}$ the interaction, which we do not even need to specify.

Within HF, one replaces the interacting Hamiltonian (1) with the non-interacting one

$$\hat{H}_*(\mathbf{k}) = \hat{H}_0(\mathbf{k}) + \hat{\Sigma}_{\rm HF}\big[\mathbf{k}, \hat{G}_*\big] \, , \tag{2}$$

which is just the quasiparticle Hamiltonian we mentioned above. In (2), $\hat{H}_0(\mathbf{k})$ is the matrix with elements $H_{0\,\alpha\beta}(\mathbf{k})$, $\hat{\Sigma}_{\rm HF}\big[\mathbf{k}, \hat{G}_*\big]$ includes only the Hartree and Fock diagrams of the self-energy functional, and

$$\hat{G}_*(i\epsilon, \mathbf{k}) = \frac{1}{i\epsilon - \hat{H}_*(\mathbf{k})} \, , \tag{3}$$

is the HF Green's function in Matsubara frequencies. RPA is a symmetry-conserving scheme consistent with HF [13], which amounts to calculate response functions using the Green's function (3) and the irreducible scattering vertex in the particle-hole channel defined as

$$\frac{\delta\hat{\Sigma}_{\rm HF}\big[\mathbf{k}, \hat{G}_*\big]}{\delta\hat{G}_*(i\epsilon, \mathbf{k}')} = \hat{f}(\mathbf{k}, \mathbf{k}') \, , \tag{4}$$

thus providing the definition of the Landau $f$-parameter tensor. Accordingly, the quasiparticle energies $\epsilon_\ell(\mathbf{k})$ and eigenfunctions $|\,\psi_\ell(\mathbf{k})\rangle$, using roman letters to distinguish them from the greek ones that label the original basis, are obtained by diagonalising

$$\begin{aligned}
H_{*\alpha\beta}(\mathbf{k}) &= H_{0\alpha\beta}(\mathbf{k}) + \frac{1}{V} \sum_{\mathbf{k}'\mu\nu} f_{\alpha\nu,\mu\beta}(\mathbf{k}, \mathbf{k}') \, \langle c^\dagger_{\nu\mathbf{k}'} c_{\mu\mathbf{k}'} \rangle \\
&= H_{0\alpha\beta}(\mathbf{k}) + \frac{T}{V} \sum_{\mathbf{k}'\mu\nu} \sum_\epsilon e^{i\epsilon 0^+} f_{\alpha\nu,\mu\beta}(\mathbf{k}, \mathbf{k}') \, G_{*\,\mu\nu}(i\epsilon, \mathbf{k}') \, ,
\end{aligned} \tag{5}$$

where $V$ is the volume, $f_{\alpha\gamma,\delta\beta}(\mathbf{k}, \mathbf{k}')$ are, through (4), the components of $\hat{f}(\mathbf{k}, \mathbf{k}')$, and the expectation values, to be determined self-consistently, are over the HF ground state. The diagonalization is accomplished by a unitary transformation $\hat{U}(\mathbf{k})$,

$$\hat{U}(\mathbf{k})^\dagger \, \hat{H}_*(\mathbf{k}) \, \hat{U}(\mathbf{k}) = \hat{\epsilon}_*(\mathbf{k}) \, ,$$

where $\hat{\epsilon}_*(\mathbf{k})$ is the diagonal matrix with elements $\epsilon_\ell(\mathbf{k})$.

By means of the Ward-Takahashi identities, Noziéres and Luttinger [14] showed that in the so-called static or $q$-limit, the transferred frequency vanishing before the transferred momentum, the fully-interacting physical current vertex multiplied by the quasiparticle residue, which we hereafter denote as $\hat{\boldsymbol{J}}^q(\mathbf{k})$, corresponds to $\boldsymbol{\nabla}_{\mathbf{k}} \hat{H}_*(\mathbf{k})$ for the quasiparticles. In the diagonal basis, which we define as HF basis, that correspondence reads

$$\begin{aligned}
\hat{\boldsymbol{J}}^q(\mathbf{k}) &\equiv \hat{U}(\mathbf{k})^\dagger \, \boldsymbol{\nabla}_{\mathbf{k}} \hat{H}_*(\mathbf{k}) \, \hat{U}(\mathbf{k}) = \hat{U}(\mathbf{k})^\dagger \, \boldsymbol{\nabla}_{\mathbf{k}} \left( \hat{U}(\mathbf{k}) \, \hat{\epsilon}_*(\mathbf{k}) \, \hat{U}(\mathbf{k})^\dagger \right) \hat{U}(\mathbf{k}) \\
&= \boldsymbol{\nabla}_{\mathbf{k}} \, \hat{\epsilon}_*(\mathbf{k}) - \left[ \hat{\epsilon}_*(\mathbf{k}), \, \hat{U}(\mathbf{k})^\dagger \, \boldsymbol{\nabla}_{\mathbf{k}} \hat{U}(\mathbf{k}) \right] \equiv \boldsymbol{\nabla}_{\mathbf{k}} \, \hat{\epsilon}_*(\mathbf{k}) + i \left[ \hat{\epsilon}_*(\mathbf{k}), \, \hat{\boldsymbol{\mathcal{A}}}^0(\mathbf{k}) \right] ,
\end{aligned} \tag{6}$$

where the first term is diagonal and the second off-diagonal in the band indices. Here, the hermitian operator $\hat{\boldsymbol{\mathcal{A}}}^{0}(\mathbf{k}) = i\,\hat{U}(\mathbf{k})^{\dagger}\,\boldsymbol{\nabla}_{\mathbf{k}}\,\hat{U}(\mathbf{k})$ is the bare Berry connection of the quasiparticles, whose matrix elements are also equal to

$$\boldsymbol{\mathcal{A}}^{0}_{\ell m}(\mathbf{k}) = i\,\langle\psi_{\ell}(\mathbf{k})\mid\boldsymbol{\nabla}_{\mathbf{k}}\,\psi_{m}(\mathbf{k})\rangle = -i\,\langle\boldsymbol{\nabla}_{\mathbf{k}}\,\psi_{\ell}(\mathbf{k})\mid\psi_{m}(\mathbf{k})\rangle\,, \tag{7}$$

through which the diagonal part of the bare Berry curvature is $\boldsymbol{\Omega}^{0}_{\ell\ell}(\mathbf{k}) = -\boldsymbol{\nabla}_{\mathbf{k}}\times\boldsymbol{\mathcal{A}}^{0}_{\ell\ell}(\mathbf{k})$.

The reason why we have reformulated Landau's Fermi-liquid theory in such more complex language is to make it clear that quasiparticles are interacting, though interaction is treated within HF plus RPA. This point becomes important when one wants to compute topological observables. As already mentioned, the first temptation is to use the eigenfunctions $\mid\psi_{n}(\mathbf{k})\rangle$ and calculate the topological invariants as one would do for non-interacting electrons, see (7). However, this procedure might be incorrect just because the quasiparticles do interact with each other.

A more rigorous approach is to directly calculate the anomalous Hall conductivity as dictated by Landau's Fermi liquid theory [14]. In Appendix A we derive the expression of the matrix elements $\sigma^{\mathrm{H}}_{ab} = -\sigma^{\mathrm{H}}_{ba}$, $a\neq b$ labelling the spatial components, of the Hall conductivity tensor, see (44), which read, explicitly,

$$
\begin{aligned}
\sigma^{\mathrm{H}}_{ab} &= -i\,\frac{e^{2}}{V}\,\sum_{\mathbf{k}}\,\sum_{\ell\neq m}\,J^{\omega}_{a\,\ell m}(\mathbf{k})\,\frac{f\big(\epsilon_{\ell}(\mathbf{k})\big)-f\big(\epsilon_{m}(\mathbf{k})\big)}{\big(\epsilon_{m}(\mathbf{k})-\epsilon_{\ell}(\mathbf{k})\big)^{2}}\,J^{\omega}_{b\,m\ell}(\mathbf{k}) \\
&= -i\,\frac{e^{2}}{V}\,\sum_{\mathbf{k}}\,\sum_{\ell\neq m}\,f\big(\epsilon_{\ell}(\mathbf{k})\big)\,\frac{J^{\omega}_{a\,\ell m}(\mathbf{k})\,J^{\omega}_{*b\,m\ell}(\mathbf{k})-J^{\omega}_{*b\,\ell m}(\mathbf{k})\,J^{\omega}_{a\,m\ell}(\mathbf{k})}{\big(\epsilon_{m}(\mathbf{k})-\epsilon_{\ell}(\mathbf{k})\big)^{2}} \\
&\equiv \frac{e^{2}}{V}\,\sum_{\mathbf{k}\ell}\,f\big(\epsilon_{\ell}(\mathbf{k})\big)\,\epsilon_{abc}\,\Omega_{c\,\ell\ell}(\mathbf{k})\,,
\end{aligned} \tag{8}
$$

where $f(x)$ is the Fermi distribution function, and

$$\boldsymbol{\Omega}_{\ell\ell}(\mathbf{k}) = -i\,\sum_{m\neq\ell}\,\frac{\boldsymbol{J}^{\omega}_{\ell m}(\mathbf{k})\times\boldsymbol{J}^{\omega}_{m\ell}(\mathbf{k})}{\big(\epsilon_{m}(\mathbf{k})-\epsilon_{\ell}(\mathbf{k})\big)^{2}}\,, \tag{9}$$

is therefore the true quasiparticle Berry curvature. The dynamic, $\omega$-limit of the current vertex, the opposite of the $q$-limit, is related to (6) through (41), specifically,

$$\boldsymbol{J}^{\omega}_{\ell m}(\mathbf{k}) = \boldsymbol{J}^{q}_{\ell m}(\mathbf{k}) - \frac{1}{V}\,\sum_{\mathbf{k}'n}\,\Gamma^{\omega}_{\ell n,nm}(\mathbf{k}',\mathbf{k})\,\boldsymbol{\nabla}_{\mathbf{k}'}\,f\big(\epsilon_{n}(\mathbf{k}')\big)\,, \tag{10}$$

where $\Gamma^{\omega}$ is the $\omega$-limit of the reducible scattering vertex (21). The term in (9) with the bare current vertices, i.e., $\Gamma^{\omega}=0$ in (10), reproduces the bare Berry curvature $\boldsymbol{\Omega}^{0}_{\ell\ell}(\mathbf{k})$, while the additional terms correspond to the desired Fermi liquid corrections, and indeed derive, see the second term in (10), from the quasiparticle Fermi surface. It is therefore reasonable to argue that also the corrections to the anomalous Hall conductivity come from the quasiparticle Fermi surface, though this conclusion seems not immediately obvious looking at (8) and (10). In the next section, we analyse a specific example and show explicitly that the above surmise is true.

# 3  A toy model calculation

We consider the Bernevig, Hughes and Zhang (BHZ) model [15] for a quantum spin-Hall insulator on a square lattice. In particular, we assume the model with full spin polarisation, only spin up bands being occupied, and at density $n = 1+\delta$, in which case the model describes a topological metal with broken time-reversal symmetry.

The Hamiltonian for the spin-up quasiparticles is assumed to be

$$
\begin{aligned}
\hat{H}_*(\mathbf{k}) &= \big(\epsilon(\mathbf{k}) - \mu\big)\,\sigma_0 + \big(M - t(\mathbf{k})\big)\,\sigma_3 + \lambda(\mathbf{k})\,\sin k_x\,\sigma_1 - \lambda(\mathbf{k})\,\sin k_y\,\sigma_2 \\
&= \big(\epsilon(\mathbf{k}) - \mu\big)\,\sigma_0 + \boldsymbol{B}(\mathbf{k})\cdot\boldsymbol{\sigma}\,,
\end{aligned} \tag{11}
$$

where $\mu$ crosses either the valence or the conduction bands, the identity, $\sigma_0$, and the Pauli matrices $\sigma_a$, $a = 1, 2, 3$, act on the orbital space, and

$$
\begin{aligned}
\boldsymbol{B}(\mathbf{k}) &= \big(\lambda(\mathbf{k})\,\sin k_x, -\lambda(\mathbf{k})\,\sin k_y, M - t(\mathbf{k})\big) \\
&= B(\mathbf{k})\,\Big(\cos\phi(\mathbf{k})\,\sin\theta(\mathbf{k}), \sin\phi(\mathbf{k})\,\sin\theta(\mathbf{k}), \cos\theta(\mathbf{k})\Big) \\
&= B(\mathbf{k})\,\Big(\sin\theta(\mathbf{k})\,\boldsymbol{v}_2(\mathbf{k}) + \cos\theta(\mathbf{k})\,\boldsymbol{v}_3(\mathbf{k})\Big)\,.
\end{aligned} \tag{12}
$$

Here, $\boldsymbol{v}_2(\mathbf{k}) = (\cos\phi(\mathbf{k}), \sin\phi(\mathbf{k}), 0)$ and $\boldsymbol{v}_3(\mathbf{k}) = (0, 0, 1)$ are orthogonal unit vectors that form a basis together with $\boldsymbol{v}_1(\mathbf{k}) = \boldsymbol{v}_2(\mathbf{k}) \times \boldsymbol{v}_3(\mathbf{k})$. This, in turn, implies that $\sigma_0$ and $\boldsymbol{v}_a(\mathbf{k})\cdot\boldsymbol{\sigma}$, $a = 1, 2, 3$, form a basis of $2 \times 2$ matrices. To simplify the notations, in what follows we use the definition $\boldsymbol{v}_0(\mathbf{k})\cdot\boldsymbol{\sigma} := \sigma_0$.

The quasiparticle Hamiltonian (11) is assumed to be invariant under inversion $\mathcal{I}$, $\mathbf{k} \to -\mathbf{k}$ and $\sigma_a \to -\sigma_a$ for $a = 1, 2$, and fourfold rotations $C_4$, $k_x \to k_y \wedge k_y \to -k_x$ and $\sigma_1 \to -\sigma_2 \wedge \sigma_2 \to \sigma_1$. This requires that the parameters $\epsilon(\mathbf{k})$, $t(\mathbf{k})$ and $\lambda(\mathbf{k})$ in (11) are invariant under both $\mathcal{I}$ and $C_4$. In addition, in order to clearly distinguish $\mu$ and $M$ from, respectively, $\epsilon(\mathbf{k})$ and $t(\mathbf{k})$, we assume that the latter average out at zero over the Brillouin zone.

The Hamiltonian $\hat{H}_*(\mathbf{k})$ is diagonalised by the unitary transformation

$$
\hat{U}(\mathbf{k}) = \mathrm{e}^{i\frac{\theta(\mathbf{k})}{2}\,\boldsymbol{v}_1(\mathbf{k})\cdot\boldsymbol{\sigma}} = \cos\frac{\theta(\mathbf{k})}{2} + i\,\sin\frac{\theta(\mathbf{k})}{2}\,\boldsymbol{v}_1(\mathbf{k})\cdot\boldsymbol{\sigma}\,,
$$

namely $\hat{U}(\mathbf{k})^\dagger\,\hat{H}_*(\mathbf{k})\,\hat{U}(\mathbf{k}) = (\epsilon(\mathbf{k}) - \mu)\,\sigma_0 + B(\mathbf{k})\,\sigma_3$. We readily find that

$$
\begin{aligned}
\hat{\boldsymbol{\mathcal{A}}}^0(\mathbf{k}) &= i\,\hat{U}(\mathbf{k})^\dagger\,\boldsymbol{\nabla}_{\mathbf{k}}\,\hat{U}(\mathbf{k}) = -\frac{\boldsymbol{\nabla}_{\mathbf{k}}\,\theta(\mathbf{k})}{2}\,\boldsymbol{v}_1(\mathbf{k})\cdot\boldsymbol{\sigma} \\
&\quad - \frac{\boldsymbol{\nabla}_{\mathbf{k}}\,\phi(\mathbf{k})}{2}\,\sin\theta(\mathbf{k})\,\boldsymbol{v}_2(\mathbf{k})\cdot\boldsymbol{\sigma} - \boldsymbol{\nabla}_{\mathbf{k}}\,\phi(\mathbf{k})\,\sin^2\frac{\theta(\mathbf{k})}{2}\,\boldsymbol{v}_3(\mathbf{k})\cdot\boldsymbol{\sigma} \\
&= \sum_{a=1}^{3}\,\mathcal{A}_a^0(\mathbf{k})\,\boldsymbol{v}_a(\mathbf{k})\cdot\boldsymbol{\sigma}\,.
\end{aligned} \tag{13}
$$

We can make a similar expansion for the currents,

$$
\hat{\boldsymbol{J}}^q(\mathbf{k}) = \sum_{a=0}^{3}\,\boldsymbol{J}_a^q(\mathbf{k})\,\boldsymbol{v}_a(\mathbf{k})\cdot\boldsymbol{\sigma}\,, \quad \hat{\boldsymbol{J}}^\omega(\mathbf{k}) = \sum_{a=0}^{3}\,\boldsymbol{J}_a^\omega(\mathbf{k})\,\boldsymbol{v}_a(\mathbf{k})\cdot\boldsymbol{\sigma}\,,
$$

where $\boldsymbol{J}_0^q(\mathbf{k}) = \boldsymbol{\nabla}_{\mathbf{k}}\,\epsilon(\mathbf{k})$, $\boldsymbol{J}_3^q(\mathbf{k}) = \boldsymbol{\nabla}_{\mathbf{k}}\,B(\mathbf{k})$ and $\boldsymbol{J}_a^q(\mathbf{k}) = 2\,B(\mathbf{k})\,\epsilon_{ab3}\,\mathcal{A}_b^0(\mathbf{k})$ for $a = 1, 2$. In the top panel of Fig. 1 we show the band structure of (11) at $\epsilon(\mathbf{k}) = 0$, $M = 1$, $t(\mathbf{k}) = \cos k_x + \cos k_y$

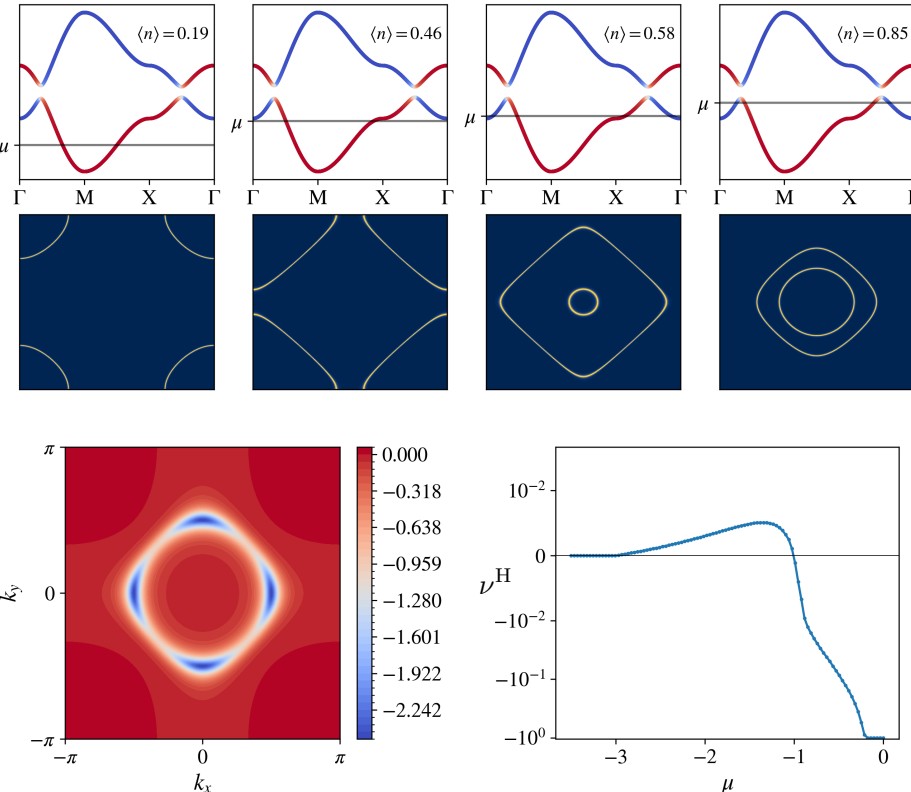

Figure 1: Top panel: band structures of (11) with $\epsilon(\mathbf{k}) = 0$, $M = 1$, $t(\mathbf{k}) = \cos k_x + \cos k_y$ and $\lambda(\mathbf{k}) = 0.2$ (top), and Fermi surfaces (bottom) for different values of $\mu$ corresponding to hole doping. Bottom panel: distribution in momentum space of the bare Berry curvature $\Omega^0(\mathbf{k})$ of the valence band (left); integral $\nu^{\mathrm{H}}$ of the bare Berry curvature in units of $2\pi$ over the occupied Fermi volume (right) as function of $\mu < 0$. When $2|\mu|$ is smaller than the gap, $\nu^{\mathrm{H}} = -1$ keeps its quantised value, otherwise it deviates reaching zero when the valence band empties, $\mu = -3$. In reality, $\nu^{\mathrm{H}}$ crosses zero also when the Fermi surface changes character, from hole-like to electron-like. We also note that $\Omega^0(\mathbf{k})$ of the valence band is peaked in magnitude at the top of the band, when the orbital character changes. This explain the fast decrease in magnitude upon hole doping.

and $\lambda(\mathbf{k}) = 0.2$, indicating the chemical potential $\mu$ for the different hole doping levels and the corresponding Fermi surfaces. In the bottom panel, we draw the momentum distribution of the bare Berry curvature of the valence band, as well as its integral in units of $2\pi$ over the occupied Fermi volume as function of the chemical potential $\mu < 0$.

We note that $\boldsymbol{v}_a(\mathbf{k}) \cdot \boldsymbol{\sigma}$, $a = 0, \ldots, 3$, are invariant under both $\mathcal{I}$ and $C_4$, and $\boldsymbol{\nabla}_{\mathbf{k}} \phi(\mathbf{k})$ as well as $\boldsymbol{\nabla}_{\mathbf{k}} \theta(\mathbf{k})$ transform as conventional vectors. Since the $f$-parameters contribute to the quasiparticle Hamiltonian (11) through the mean-field decoupling, the expression of $\boldsymbol{B}(\mathbf{k})$ in (12) implies that the most general $\hat{f}$-tensor in the original basis must be of the form

$$\hat{f}(\mathbf{k}, \mathbf{k}') = \sum_{a,b=0,2,3} f_{ab}(\mathbf{k}, \mathbf{k}') \left(\boldsymbol{v}_a(\mathbf{k}) \cdot \boldsymbol{\sigma}\right) \otimes \left(\boldsymbol{v}_b(\mathbf{k}') \cdot \boldsymbol{\sigma}\right), \tag{14}$$

thus lacking terms that involve $\boldsymbol{v}_1(\mathbf{k}) \cdot \boldsymbol{\sigma}$, and where symmetry only requires that the expansion coefficients are invariant under $\mathcal{I}$ and $C_4$. Upon rotation in the HF basis, (14) maintains the same form with modified coefficients, which we still denote as $f_{ab}(\mathbf{k}, \mathbf{k}')$ for sake of simplicity. We can further identify two distinct tensors, $\hat{f}^0(\mathbf{k}, \mathbf{k}') = \hat{f}^0(-\mathbf{k}, \mathbf{k}') = \hat{f}^0(\mathbf{k}, -\mathbf{k}')$ and $\hat{f}^1(\mathbf{k}, \mathbf{k}') = -\hat{f}^1(-\mathbf{k}, \mathbf{k}') = -\hat{f}^1(\mathbf{k}, -\mathbf{k}')$, using the same notation as in conventional single-band Fermi liquids [16]. It is precisely $\hat{f}^1(\mathbf{k}, \mathbf{k}')$ that may contribute to (10), which, without loss of generality, can be taken in the HF basis of the form

$$\hat{f}^1(\mathbf{k}, \mathbf{k}') = \sum_{a,b=0,2,3} f_{ab}^1(\mathbf{k}, \mathbf{k}') \, \boldsymbol{J}_a^q(\mathbf{k}) \cdot \boldsymbol{J}_b^q(\mathbf{k}') \left(\boldsymbol{v}_a(\mathbf{k}) \cdot \boldsymbol{\sigma}\right) \otimes \left(\boldsymbol{v}_b(\mathbf{k}') \cdot \boldsymbol{\sigma}\right),$$

where, for consistency, $f_{ab}^1(\mathbf{k}, \mathbf{k}') = f_{ab}^1(-\mathbf{k}, \mathbf{k}') = f_{ab}^1(\mathbf{k}, -\mathbf{k}')$ vary in momentum space orthogonally to the corresponding currents, i.e.,

$$\boldsymbol{J}_a^q(\mathbf{k}) \cdot \boldsymbol{\nabla}_{\mathbf{k}} f_{ab}^1(\mathbf{k}, \mathbf{k}') = \boldsymbol{\nabla}_{\mathbf{k}'} f_{ab}^1(\mathbf{k}, \mathbf{k}') \cdot \boldsymbol{J}_b^q(\mathbf{k}') = 0, \tag{15}$$

for $a, b = 0, 2, 3$. In reality, only the components $f_{ab}^1(\mathbf{k}, \mathbf{k}')$, $a = 0, 2, 3$ and $b = 0, 3$, contribute to (10), in which case (21) implies that $\Gamma_{ab}^{1\omega}(\mathbf{k}, \mathbf{k}') \equiv f_{ab}^1(\mathbf{k}, \mathbf{k}')$. It is worth remarking that $\hat{f}(\mathbf{k}, \mathbf{k}')$ in (14) contributes to $\hat{H}_*(\mathbf{k})$ in (11) through the HF self-energy, see (5). Therefore, if we reasonably assume that the HF ground state does not carry any current, then the mean-field terms generated by $\hat{f}^1(\mathbf{k}, \mathbf{k}')$ must vanish at self-consistency, which requires, at the very least, non-singular coefficients $f_{ab}^1(\mathbf{k}, \mathbf{k}')$. This observation will be useful later on.

We further define the diagonal matrix $\hat{\mathcal{P}}(\mathbf{k})$ with elements $f\left(\epsilon_\ell(\mathbf{k})\right)$, which we can write as

$$\hat{\mathcal{P}}(\mathbf{k}) = \frac{f\left(\epsilon_1(\mathbf{k})\right) + f\left(\epsilon_2(\mathbf{k})\right)}{2} \, \boldsymbol{v}_0(\mathbf{k}) \cdot \boldsymbol{\sigma} + \frac{f\left(\epsilon_1(\mathbf{k})\right) - f\left(\epsilon_2(\mathbf{k})\right)}{2} \, \boldsymbol{v}_3(\mathbf{k}) \cdot \boldsymbol{\sigma}$$
$$\equiv \mathcal{P}_0(\mathbf{k}) \, \boldsymbol{v}_0(\mathbf{k}) \cdot \boldsymbol{\sigma} + \mathcal{P}_3(\mathbf{k}) \, \boldsymbol{v}_3(\mathbf{k}) \cdot \boldsymbol{\sigma},$$

where the conduction band 1 corresponds to $\sigma_3 = +1$ with energy $\epsilon_1(\mathbf{k}) = \epsilon(\mathbf{k}) - \mu + B(\mathbf{k})$, and the valence band 2 to $\sigma_3 = -1$ and $\epsilon_2(\mathbf{k}) = \epsilon(\mathbf{k}) - \mu - B(\mathbf{k})$. With those definitions, (10) transforms into an equation for each component in the matrix basis, specifically, and

exploiting the spatial symmetries,

$$
\begin{aligned}
\boldsymbol{J}_a^\omega(\mathbf{k}) &= \boldsymbol{J}_a^q(\mathbf{k}) - \left(1 - \delta_{a,1}\right) \frac{2}{V} \sum_{\mathbf{k}'} \sum_{b=0,3} f_{ab}^1(\mathbf{k}, \mathbf{k}') \, \boldsymbol{\nabla}_{\mathbf{k}'} \, \mathcal{P}_b(\mathbf{k}') \left( \boldsymbol{J}_a^q(\mathbf{k}) \cdot \boldsymbol{J}_b^q(\mathbf{k}') \right) \\
&= \boldsymbol{J}_a^q(\mathbf{k}) \left\{ 1 - \left(1 - \delta_{a,1}\right) \frac{1}{V} \sum_{\mathbf{k}'} \sum_{b=0,3} f_{ab}^1(\mathbf{k}, \mathbf{k}') \, \boldsymbol{\nabla}_{\mathbf{k}'} \, \mathcal{P}_b(\mathbf{k}') \cdot \boldsymbol{J}_b^q(\mathbf{k}') \right\} \\
&\equiv \boldsymbol{J}_a^q(\mathbf{k}) \left( 1 + \left(1 - \delta_{a,1}\right) \frac{F_a^1(\mathbf{k})}{2} \right).
\end{aligned}
\tag{16}
$$

The anomalous Hall conductivity $\sigma^{\mathrm{H}} = \sigma_{xy}^{\mathrm{H}}$ in (8) can be simply written as

$$
\begin{aligned}
\sigma^{\mathrm{H}} &= -i \frac{e^2}{V} \sum_{\mathbf{k}} \sum_{a,b=1}^{2} \frac{J_{xa}^\omega(\mathbf{k}) \, J_{yb}^\omega(\mathbf{k})}{4B(\mathbf{k})^2} \, \mathrm{Tr}\left( \left[ \mathcal{P}_3(\mathbf{k}) \, \boldsymbol{v}_3(\mathbf{k}) \cdot \boldsymbol{\sigma}, \boldsymbol{v}_a(\mathbf{k}) \cdot \boldsymbol{\sigma} \right] \boldsymbol{v}_b(\mathbf{k}) \cdot \boldsymbol{\sigma} \right) \\
&= \frac{e^2}{2V} \sum_{\mathbf{k}} \sum_{a,b=1}^{2} \frac{\mathcal{P}_3(\mathbf{k})}{B(\mathbf{k})^2} \, \epsilon_{ab3} \, \boldsymbol{J}_a^\omega(\mathbf{k}) \times \boldsymbol{J}_b^\omega(\mathbf{k}) \cdot \boldsymbol{v}_3(\mathbf{k}) \\
&= -\frac{e^2}{V} \sum_{\mathbf{k}} \mathcal{P}_3(\mathbf{k}) \left( 1 + \frac{F_2^1(\mathbf{k})}{2} \right) \boldsymbol{\nabla}_{\mathbf{k}} \times \left( \cos\theta(\mathbf{k}) \, \boldsymbol{\nabla}_{\mathbf{k}} \, \phi(\mathbf{k}) \right) \cdot \boldsymbol{v}_3(\mathbf{k}) \\
&= \frac{e^2}{V} \sum_{\mathbf{k}} \cos\theta(\mathbf{k}) \left( 1 + \frac{F_2^1(\mathbf{k})}{2} \right) \boldsymbol{\nabla}_{\mathbf{k}} \, \mathcal{P}_3(\mathbf{k}) \times \boldsymbol{\nabla}_{\mathbf{k}} \, \phi(\mathbf{k}) \cdot \boldsymbol{v}_3(\mathbf{k}) \\
&\quad + \frac{e^2}{2V} \sum_{\mathbf{k}} \cos\theta(\mathbf{k}) \, \mathcal{P}_3(\mathbf{k}) \, \boldsymbol{\nabla}_{\mathbf{k}} \, F_2^1(\mathbf{k}) \times \boldsymbol{\nabla}_{\mathbf{k}} \, \phi(\mathbf{k}) \cdot \boldsymbol{v}_3(\mathbf{k}).
\end{aligned}
\tag{17}
$$

We note that the first term in the last equality of (17) is a genuine Fermi surface contribution. Concerning the last term, we recall that $\boldsymbol{J}_2^q(\mathbf{k}) \propto \boldsymbol{\mathcal{A}}_1^0(\mathbf{k}) \propto \boldsymbol{\nabla}_{\mathbf{k}} \, \theta(\mathbf{k})$ and, because of (15), the vector product $\boldsymbol{\nabla}_{\mathbf{k}} \, F_2^1(\mathbf{k}) \times \boldsymbol{\nabla}_{\mathbf{k}} \, \phi(\mathbf{k})$ is proportional to $\sin k_x \sin k_y$, odd under $C_4$, which therefore averages out at zero upon summing over $\mathbf{k}$ since $\mathcal{P}_3(\mathbf{k})$ and $\cos\theta(\mathbf{k})$ are both invariant. In conclusion

$$
\sigma^{\mathrm{H}} = \frac{e^2}{V} \sum_{\mathbf{k}} \left( 1 + \frac{F_2^1(\mathbf{k})}{2} \right) \cos\theta(\mathbf{k}) \, \boldsymbol{\nabla}_{\mathbf{k}} \, \mathcal{P}_3(\mathbf{k}) \times \boldsymbol{\nabla}_{\mathbf{k}} \, \phi(\mathbf{k}) \cdot \boldsymbol{v}_3(\mathbf{k}) \equiv \sigma_0^{\mathrm{H}} \left( 1 + \frac{F_2^1}{2} \right),
\tag{18}
$$

where $\sigma_0^{\mathrm{H}}$ is the bare value, i.e., neglecting the vertex corrections, and $F_2^1$ is a weighted average over the Fermi surfaces. Therefore, in the model (11) we can explicitly verify that the corrections to the anomalous Hall conductivity only derive from the quasiparticle Fermi surface, as we earlier conjectured. We believe that this result has a more general validity. For completeness, the Drude weight $D_{xx} = D_{yy} = D$ can be readily found through (40),

$$
D = D_0 \left( 1 + \frac{F^1}{2} \right),
\tag{19}
$$

where

$$
D_0 = -\frac{e^2}{2V} \sum_{\mathbf{k}} \sum_{\ell=1}^{2} \frac{\partial f\left(\epsilon_\ell(\mathbf{k})\right)}{\partial \epsilon_\ell(\mathbf{k})} \, \boldsymbol{\nabla}_{\mathbf{k}} \, \epsilon_\ell(\mathbf{k}) \cdot \boldsymbol{\nabla}_{\mathbf{k}} \, \epsilon_\ell(\mathbf{k}) = \frac{e^2}{2V} \sum_{\mathbf{k}} \sum_{\ell=1}^{2} f\left(\epsilon_\ell(\mathbf{k})\right) \nabla_{\mathbf{k}}^2 \, \epsilon_\ell(\mathbf{k}),
\tag{20}
$$

is the bare value, and $F^1$ is again a weighted average over the Fermi surfaces of $F_0^1(\mathbf{k})$ and $F_3^1(\mathbf{k})$, see (16). In this case, $F^1/2$ is the correction to the optical mass with respect to the quasiparticle effective one, defined through the last equation in (20).

We emphasise that the Fermi liquid corrections to both anomalous Hall conductivity (18) and Drude weight (19) stem from the fact that we are studying the response to a uniform electric field, which has to be evaluated in the $\omega$-limit. As a consequence, the dressed current vertex that enters the response functions is $\boldsymbol{J}^\omega$ instead of $\boldsymbol{J}^q$, the latter being the one obtainable from the Ward-Takahashi identity. In a Fermi liquid, $\boldsymbol{J}^\omega \neq \boldsymbol{J}^q$ because the quasiparticle Fermi surface induces a non-analyticity at $\omega = q = 0$, which is the key to the microscopic derivation of Landau's Fermi liquid theory [14].

To conclude, we mention that, if instead of a fully spin-polarised BHZ model as in (11), we considered the same model without explicitly breaking time-reversal symmetry, or models sharing similar topological ingredients, we could still discuss topological properties as the non-quantised intrinsic component of the spin Hall conductivity, see, e.g., [17, 18].

## 4   Conclusions

Landau's theory of topological Fermi liquids predicts that the residual interactions among the quasiparticles, the $f$-parameters, yield corrections not only to conventional thermodynamic susceptibilities and longitudinal transport coefficients, like the Drude weight, but also to the intrinsic anomalous Hall conductivity. The latter is therefore not expressible solely in terms of the Berry phases acquired by the quasiparticle Bloch states adiabatically evolving on the Fermi surface, as one would intuitively argue [1].

This observation is the metallic counterpart of recent results [10, 11] showing that the Chern number of two-dimensional quantum anomalous Hall insulators not necessarily coincides with the topological invariant [19, 20, 21, 22] corresponding to the winding number $W(G)$, also denoted as $N_3(G)$, of the map $(\epsilon, \mathbf{k}) \to \hat{G}(i\epsilon, \mathbf{k}) \in \mathrm{GL}(n, \mathbb{C})$, where $\hat{G}(i\epsilon, \mathbf{k})$ is the fully-interacting Green's function. Indeed, as demonstrated in [10], this winding number is equivalent to that of the map $(\epsilon, \mathbf{k}) \to \hat{G}_*(i\epsilon, \mathbf{k})$, where $\hat{G}_*(i\epsilon, \mathbf{k})$ is obtained by filtering out from $\hat{G}(i\epsilon, \mathbf{k})$ the quasiparticle residue, see Appendix B, and thus coincides with the quasiparticle Green's function (3) in the doped insulator. Correspondingly, $W(G_*)$ reduces upon doping to the non-quantised $\sigma_0^{\mathrm{H}}$ in units of $e^2/2\pi$, and the corrections predicted in [10, 11] to the Fermi liquid ones we have just derived. We remark that, since $\hat{G}_*(i\epsilon, \mathbf{k})$ in the doped insulator has singular eigenvalues at $\epsilon = 0$ on the quasiparticle Fermi surface, $W(G_*)$ is, strictly speaking, not anymore a genuine winding number, though it remains perfectly defined. With this proviso, we hereafter refer to $W(G_*)$ still as the winding number.

In the following, we show that, elaborating on the previous observation, one can draw interesting and rather general conclusions. We start emphasising that Refs. [10] and [11] rationalise the puzzling evidence of Mott insulators with, presumably [23], vanishing Chern number and yet finite winding number due to the existence of in-gap topological bands of Green's function zeros [24, 25, 26, 27, 28, 29]. It is reasonable to conjecture that doping such Mott insulators with $\sigma^{\mathrm{H}} = 0$ but $W(G_*) \propto \sigma_0^{\mathrm{H}} \neq 0$ may lead, without any change of symmetry, to topological metals with finite $\sigma^{\mathrm{H}}$. For simplicity, we assume that such metal can be described as in Sect. 3, and use the results of that section to infer how it may continuously transform into the Mott

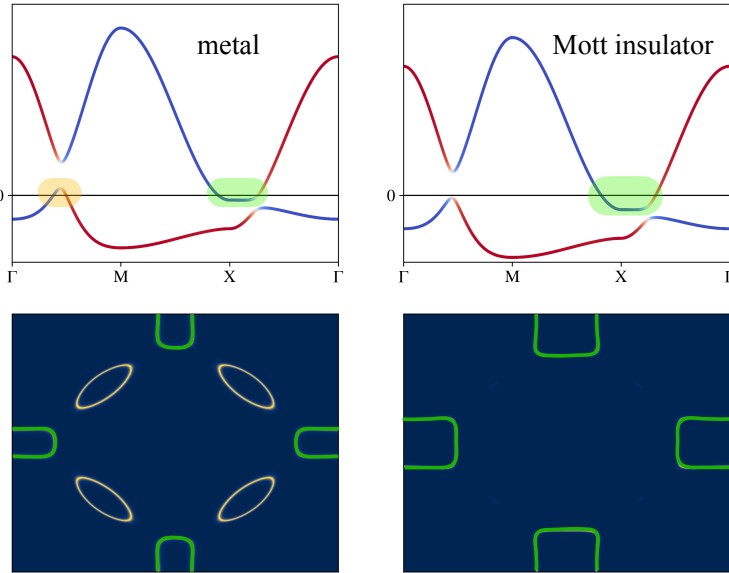

Figure 2: Left panels: hypothetical quasiparticle band structure in the metal phase (top) showing two distinct sets of quasiparticle Fermi pockets (bottom), one representing physical Fermi pockets, in yellow, which account for the hole doping, and the other, in green, being Luttinger pockets. Right panels: the same as left ones but on the Mott insulating side at $\delta = 0$. Here, only the Luttinger pockets exist.

insulator as the doping $\delta \to 0$.

The first and most obvious temptation is to assume that $\delta \to 0$ simply corresponds to a quasiparticle Fermi surface that shrinks into a point and disappears in the Mott insulator. In that case, since the bare $\sigma_0^{\mathrm{H}}$ is expected to reach its quantised maximum magnitude when the valence band is full and the conduction one empty, the only possibility for $\sigma^{\mathrm{H}}$ in (18) to vanish is that $F_2^1 \to -2$ for $\delta \to 0$. However, from the definition of $F_2^1(\mathbf{k})$ in (16), we would rather conclude that $F_2^1 \to 0$ for $\delta \to 0$, unless $f_{2a}^1(\mathbf{k}, \mathbf{k}')$, $a = 0, 3$, diverge sufficiently fast to compensate the vanishing Fermi volume. As we earlier discussed, such possibility has to be discarded, which implies that the assumption of a quasiparticle Fermi surface that disappears when $\delta \to 0$ is not consistent with $\sigma^{\mathrm{H}} \to 0$, hence that a quasiparticle Fermi surface does survive till $\delta = 0$ to enforce $F_2^1 \to -2$. Since quasiparticle Fermi surfaces, poles of $\det\big(\hat{G}_*(0, \mathbf{k})\big)$, comprise both physical Fermi and Luttinger surfaces [30, 31], poles and zeros of $\det\big(\hat{G}(0, \mathbf{k})\big)$, respectively, see also Appendix B, we must conclude that the quasiparticle Fermi surface that exists at the transition into the Mott insulator is actually a Luttinger surface, which is allowed also in Mott insulators. This, in turn, implies that one of the in-gap bands of zeros of the Mott insulator right after the transition must cross the chemical potential and thus form a Luttinger surface, which is the smooth evolution of that one in the weakly doped metal phase. In Fig. 2 we show a hypothetical quasiparticle band structure that realises the above scenario, obtained by properly tuning the parameters in (11). The metal phase, left panels, exhibits both Fermi pockets, in yellow, which account just for the hole doping [32], and Luttinger ones, in green. Only the latter remains in the Mott insulator at $\delta = 0$, right panels.

This physical scenario, which we have unveiled by quite general arguments, is fully consistent with the results of [10, 11], since it predicts a Luttinger surface in the Mott insulator which is known to yield a violation of Luttinger's theorem [33, 32]. Such violation directly explains, by means of the Streda formula [34, 35], why in the insulator $\sigma^{\mathrm{H}}$ can vanish despite $W(G) \neq 0$. We also mention that recent numerical simulations of model interacting topological insulators [36, 29] find evidence of bands of Green's function zeros in the topological phase before the Mott transition, and which, should they cross the chemical potential and form a Luttinger surface, could provide a simple explanation [36] for the intriguing properties of $SmB_6$ and $YbB_{12}$ topological Kondo insulators.

If we take for granted the existence of a Luttinger surface in the topological metal at $\delta \ll 1$, we can draw a further conclusion. When $\delta \to 0$, the Drude weight (19) must vanish. However, $D_0$ in (20) is finite for $\delta \to 0$ because of the Luttinger surface, which implies that $D \to 0$ because $F^1 \to -2$. In other words, while the quasiparticle effective mass, defined by the last equality in (20), is smooth approaching the Mott phase, the optical mass diverges. Remarkably, this is precisely the scenario that Grilli and Kotliar uncovered [37] in the $t - J$ model by a large-$N$ expansion around the saddle point within the slave-boson formalism. The analogy suggests that also in the weakly doped $t - J$ model there must be a Luttinger surface, and, possibly, also in other weakly doped Mott insulators, irrespective whether topology is involved. The advantage of the latter is that it allows straight reaching this conclusion because the winding number, a topological property of the Green's function, can well be finite in both metal and insulator.

# A   Precise definition of the Fermi-liquid response functions

The correspondence between the low frequency, long wavelength and low temperature physical response functions and those obtained from the Hamiltonian (1) treated by HF+RPA is in reality not straight. Indeed, that correspondence can be drawn only in the limits where one can make use of the Ward-Takahashi identities [14], therefore only in connection with densities associated to conserved quantities and their corresponding currents, and either in the $q$-limit or in the $\omega$-limit depending on the specific response function.
This point is particularly important in multiband models and for the current-current response function. Let us therefore begin by discussing the latter [14, 13]. The Bethe-Salpeter equation for the reducible vertex in terms of the $f$-parameters is

$$
\begin{aligned}
\Gamma_{\ell m, np}(\mathbf{k}, \mathbf{k}'; i\omega, \mathbf{q}) = {}& f_{\ell m, np}(\mathbf{k}, \mathbf{k}') \\
& + \frac{1}{V} \sum_{\mathbf{p}} \sum_{qs} f_{\ell s, qp}(\mathbf{k}, \mathbf{p}) \, R_{sq}(\mathbf{p}; i\omega, \mathbf{q}) \, \Gamma_{qm, ns}(\mathbf{p}, \mathbf{k}'; i\omega, \mathbf{q}) ,
\end{aligned} \tag{21}
$$

where $\omega$ and $\mathbf{q}$ are, respectively, the frequency and momentum transferred in the particle-hole channel, and the kernel

$$
\begin{aligned}
R_{sq}(\mathbf{p}; i\omega, \mathbf{q}) &= T \sum_{\epsilon} \frac{1}{i\epsilon + i\omega - \epsilon_q(\mathbf{p} + \mathbf{q})} \, \frac{1}{i\epsilon - \epsilon_s(\mathbf{p})} \\
&= \frac{f\big(\epsilon_s(\mathbf{p})\big) - f\big(\epsilon_q(\mathbf{p} + \mathbf{q})\big)}{i\omega + \epsilon_s(\mathbf{p}) - \epsilon_q(\mathbf{p} + \mathbf{q})} .
\end{aligned} \tag{22}
$$

Equation (21) can be shortly written as

$$\Gamma = f + f \odot R \odot \Gamma = f + \Gamma \odot R \odot f, \tag{23}$$

where the symbol $\odot$ denotes sum of internal indices and momenta. Hereafter, we make a series of formal and exact manipulations of the Bethe-Salpeter equation similar to those exploited by Noziéres and Luttinger [14] to derive Landau's Fermi liquid theory. We decided to show them explicitly since they might be not familiar to everybody.
From (23) we get

$$f = \Gamma \odot \left(1 + R \odot \Gamma\right)^{-1}. \tag{24}$$

We can take the $q$-limit of (23), sending first $\omega \to 0$ and then $\mathbf{q} \to \mathbf{0}$,

$$\Gamma^q = f + \Gamma^q \odot R^q \odot f = f + f \odot R^q \odot \Gamma, \tag{25}$$

and find

$$f = \left(1 + \Gamma^q \odot R^q\right)^{-1} \odot \Gamma^q. \tag{26}$$

Comparing (24) with (26) we obtain

$$\Gamma \left(1 + R \odot \Gamma\right)^{-1} = \left(1 + \Gamma^q \odot R^q\right)^{-1} \Gamma^q,$$

which implies

$$\left(1 + \Gamma^q \odot R^q\right) \Gamma = \Gamma^q \left(1 + R \odot \Gamma\right),$$

hence

$$\Gamma = \Gamma^q + \Gamma^q \odot \left(R - R^q\right) \odot \Gamma = \Gamma^q + \Gamma \odot \left(R - R^q\right) \odot \Gamma^q. \tag{27}$$

On the other hand, from (27) it follows that

$$\Gamma^q = \Gamma \odot \left[1 + \left(R - R^q\right) \odot \Gamma\right]^{-1},$$

and so,

$$\begin{aligned}
1 + R^q \odot \Gamma^q &= 1 + R^q \odot \Gamma \odot \left[1 + \left(R - R^q\right) \odot \Gamma\right]^{-1} \\
&= \left[1 + \left(R - R^q\right) \odot \Gamma\right] \odot \left[1 + \left(R - R^q\right) \odot \Gamma\right]^{-1} \\
&\quad + R^q \odot \Gamma \odot \left[1 + \left(R - R^q\right) \odot \Gamma\right]^{-1} \\
&= \left(1 + R \odot \Gamma\right) \odot \left[1 + \left(R - R^q\right) \odot \Gamma\right]^{-1}.
\end{aligned} \tag{28}$$

The dressed current vertex $\boldsymbol{J}$ is obtained from the bare one $\boldsymbol{J}_0$, which we actually do not know, through the Bethe-Salpeter equation

$$\boldsymbol{J} = \boldsymbol{J}_0 + \boldsymbol{J}_0 \odot R \odot \Gamma = \boldsymbol{J}_0 + \Gamma \odot R \odot \boldsymbol{J}_0, \tag{29}$$

whose $q$-limit is $\boldsymbol{J}^q = \boldsymbol{J}_0 + \boldsymbol{J}_0 \odot R^q \odot \Gamma^q$. As before, we can use (29) to solve for the unknown $\boldsymbol{J}_0$,

$$\boldsymbol{J}_0 = \boldsymbol{J} \odot \left(1 + R \odot \Gamma\right)^{-1}, \tag{30}$$

and, taking the $q$-limit as well as making use of (28),

$$
\begin{aligned}
\boldsymbol{J}_0 &= \boldsymbol{J}^q \odot \left(1 + R^q \odot \Gamma^q\right)^{-1} \\
&= \boldsymbol{J}^q \odot \left[1 + \left(R - R^q\right) \odot \Gamma\right] \odot \left(1 + R \odot \Gamma\right)^{-1} \\
&= \left(1 + \Gamma \odot R\right)^{-1} \odot \left[1 + \Gamma \odot \left(R - R^q\right)\right] \odot \boldsymbol{J}^q \,,
\end{aligned}
\tag{31}
$$

which, compared with (30), leads to

$$
\boldsymbol{J} = \boldsymbol{J}^q + \boldsymbol{J}^q \odot \left(R - R^q\right) \odot \Gamma = \boldsymbol{J}^q + \Gamma \odot \left(R - R^q\right) \odot \boldsymbol{J}^q \,.
\tag{32}
$$

The current-current response function is defined, still in short notations, through

$$
\chi_{\boldsymbol{JJ}} \equiv \mathrm{Tr}\left(\boldsymbol{J} \odot R \odot \boldsymbol{J}_0\right) = \mathrm{Tr}\left(\boldsymbol{J}_0 \odot R \odot \boldsymbol{J}\right),
\tag{33}
$$

whose $q$-limit is therefore $\chi^q = \mathrm{Tr}\left(\boldsymbol{J}^q \odot R^q \odot \boldsymbol{J}_0\right)$. We can thus write, making use of (29) and (32),

$$
\begin{aligned}
\chi_{\boldsymbol{JJ}} &= \chi^q + \mathrm{Tr}\left(\boldsymbol{J} \odot R \odot \boldsymbol{J}_0\right) - \mathrm{Tr}\left(\boldsymbol{J}^q \odot R^q \odot \boldsymbol{J}_0\right) \\
&= \chi^q + \mathrm{Tr}\left(\boldsymbol{J}^q \odot \left(R - R^q\right) \odot \boldsymbol{J}_0\right) + \mathrm{Tr}\left(\boldsymbol{J}^q \odot \left(R - R^q\right) \odot \Gamma \odot R \odot \boldsymbol{J}_0\right) \\
&= \chi^q + \mathrm{Tr}\left(\boldsymbol{J}^q \odot \Delta \odot \boldsymbol{J}\right) \\
&= \chi^q + \mathrm{Tr}\left(\boldsymbol{J}^q \odot \Delta \odot \boldsymbol{J}^q\right) + \mathrm{Tr}\left(\boldsymbol{J}^q \odot \Delta \odot \Gamma \odot \Delta \odot \boldsymbol{J}^q\right),
\end{aligned}
\tag{34}
$$

where we have defined $\Delta = R - R^q$. We can now finally make use of symmetries and of the Ward-Takahashi identities. We observe that gauge symmetry entails that $\chi^q$ must precisely cancel the diamagnetic term, which we also do not know, and the Ward-Takahashi identities that $\boldsymbol{J}^q(\mathbf{k}) \equiv \boldsymbol{\nabla}_{\mathbf{k}} \hat{H}_*(\mathbf{k})$ [14]. Correspondingly, the component $\sigma_{ab}(i\omega, \mathbf{q})$ of the conductivity tensor reads

$$
\sigma_{ab} = i\, e^2 \lim_{\omega \to 0} \lim_{\mathbf{q} \to \mathbf{0}} \frac{1}{i\omega} \left\{ \mathrm{Tr}\left(J_a^q \odot \Delta \odot J_b^q\right) + \mathrm{Tr}\left(J_a^q \odot \Delta \odot \Gamma \odot \Delta \odot J_b^q\right) \right\},
\tag{35}
$$

thus involving the $\omega$-limit, first $\mathbf{q} \to \mathbf{0}$ and then $\omega \to 0$.

It is worth noticing that in (34) the kernel $R$ is substituted by $\Delta = R - R^q$. In other words, we have been obliged to manipulate RPA in order to build a correspondence with the physical response function and get rid of the unknown bare current vertex and diamagnetic term. From the explicit expression of $R$ in (22), we find that

$$
\begin{aligned}
\Delta_{\ell m}(\mathbf{k}; \omega, \mathbf{q}) &= R_{\ell m}(\mathbf{k}; \omega, \mathbf{q}) - R_{\ell m}^q(\mathbf{k}) \\
&= \frac{f\left(\epsilon_m(\mathbf{k})\right) - f\left(\epsilon_\ell(\mathbf{k} + \mathbf{q})\right)}{i\omega + \left(\epsilon_m(\mathbf{k}) - \epsilon_\ell(\mathbf{k} + \mathbf{q})\right)} - \lim_{\mathbf{q} \to \mathbf{0}} \frac{f\left(\epsilon_m(\mathbf{k})\right) - f\left(\epsilon_\ell(\mathbf{k} + \mathbf{q})\right)}{\epsilon_m(\mathbf{k}) - \epsilon_\ell(\mathbf{k} + \mathbf{q})} \,.
\end{aligned}
\tag{36}
$$

Recalling that we are in any case interested in small $\mathbf{q}$ and $\omega$, then (36) for $\ell = m$ is at leading order, and defining $\boldsymbol{v}_\ell(\mathbf{k}) = \boldsymbol{\nabla}_{\mathbf{k}} \epsilon_\ell(\mathbf{k})$ the group velocity,

$$
\Delta_{\ell\ell}(\mathbf{k}; \omega, \mathbf{q}) \simeq -\frac{\partial f\left(\epsilon_\ell(\mathbf{k})\right)}{\partial \epsilon_\ell(\mathbf{k})} \frac{i\omega}{i\omega - \boldsymbol{v}_\ell(\mathbf{k}) \cdot \mathbf{q}} \,,
\tag{37}
$$

which is the standard result in the single-band case [14]. On the contrary, for $\ell \neq m$ and assuming $\epsilon_\ell(\mathbf{k}) \neq \epsilon_m(\mathbf{k})$, we readily find that

$$\Delta_{\ell m}(\mathbf{k}; \omega, \mathbf{q}) \simeq -i\omega \, \frac{f(\epsilon_m(\mathbf{k})) - f(\epsilon_\ell(\mathbf{k}))}{(\epsilon_\ell(\mathbf{k}) - \epsilon_m(\mathbf{k}))^2} \,, \tag{38}$$

which vanishes linearly in $\omega$. Since the $\omega$-limit appears in the conductivity, then

$$\begin{aligned}
\Delta_{\ell m}^\omega(\mathbf{k}) &= \lim_{\omega \to 0} \lim_{\mathbf{q} \to \mathbf{0}} \Delta_{\ell m}(\mathbf{k}, \omega, \mathbf{q}) = -\delta_{m\ell} \, \frac{\partial f(\epsilon_\ell(\mathbf{k}))}{\partial \epsilon_\ell(\mathbf{k})} \,, \\
\dot{\Delta}_{\ell m}^\omega(\mathbf{k}) &= \lim_{\omega \to 0} \lim_{\mathbf{q} \to \mathbf{0}} \frac{\partial \Delta_{\ell m}(\mathbf{k}, \omega, \mathbf{q})}{\partial i\omega} = -(1 - \delta_{m\ell}) \, \frac{f(\epsilon_m(\mathbf{k})) - f(\epsilon_\ell(\mathbf{k}))}{(\epsilon_\ell(\mathbf{k}) - \epsilon_m(\mathbf{k}))^2} \,.
\end{aligned} \tag{39}$$

In other words, the matrix $\Delta^\omega$ is diagonal while its derivative $\dot{\Delta}^\omega$ with respect to $\omega$ and calculated at $\omega = 0$ is off-diagonal. This result is important in the calculation of the Hall conductivity.

However, let us at first calculate the longitudinal conductivity, i.e., the variation of the electric current in the same direction, e.g., $x$, of a uniform and static electric field, divided by the field strength. From (35), moving to the real frequency axis, $i\omega \to \omega + i0^+$ with small $\omega$ and setting $\mathbf{q} = \mathbf{0}$, we find

$$\begin{aligned}
\sigma_{xx}(\omega) &= i \, \frac{e^2}{\omega + i0^+} \left\{ \mathrm{Tr}\left( J_x^q \odot \Delta^\omega \odot J_x^q \right) + \mathrm{Tr}\left( J_x^q \odot \Delta^\omega \odot \Gamma^\omega \odot \Delta^\omega \odot J_x^q \right) \right\} \\
&= i \, \frac{1}{\omega + i0^+} \, e^2 \, \mathrm{Tr}\left( J_x^q \odot \Delta^\omega \odot J_x^\omega \right) \equiv i \, \frac{1}{\omega + i0^+} \, D_{xx} \,,
\end{aligned} \tag{40}$$

where $D_{xx}$ is the Drude weight and $\boldsymbol{J}^\omega$ the $\omega$-limit of the current vertex that, through (32), satisfies

$$\boldsymbol{J}^\omega = \boldsymbol{J}^q + \boldsymbol{J}^q \odot \Delta^\omega \odot \Gamma^\omega = \boldsymbol{J}^q + \Gamma^\omega \odot \Delta^\omega \odot \boldsymbol{J}^q \,. \tag{41}$$

Let us now calculate the Hall conductivity through the matrix elements $\sigma_{ab}$ of (35). For that we can simply adapt the results in [10], which imply that the antisymmetric combination $(\sigma_{ab} - \sigma_{ba})/2$ defines the element $\sigma_{ab}^{\mathrm{H}}$ of the anomalous Hall conductivity, which can be shown are simply

$$\sigma_{ab}^{\mathrm{H}} = i \, e^2 \lim_{\omega \to 0} \lim_{\mathbf{q} \to \mathbf{0}} \frac{\partial}{\partial i\omega} \left\{ \mathrm{Tr}\left( J_a^q \odot \Delta \odot J_b^q \right) + \mathrm{Tr}\left( J_a^q \odot \Delta \odot \Gamma \odot \Delta \odot J_b^q \right) \right\} \,. \tag{42}$$

Therefore,

$$\begin{aligned}
\sigma_{ab}^{\mathrm{H}} = i \, e^2 \Big\{ &\mathrm{Tr}\left( J_a^q \odot \dot{\Delta}^\omega \odot J_b^q \right) + \mathrm{Tr}\left( J_a^q \odot \dot{\Delta}^\omega \odot \Gamma^\omega \odot \Delta^\omega \odot J_b^q \right) \\
&+ \mathrm{Tr}\left( J_a^q \odot \Delta^\omega \odot \Gamma^\omega \odot \dot{\Delta}^\omega \odot J_b^q \right) + \mathrm{Tr}\left( J_a^q \odot \Delta^\omega \odot \dot{\Gamma}^\omega \odot \Delta^\omega \odot J_b^q \right) \Big\} \,,
\end{aligned} \tag{43}$$

involving $\Delta^\omega$ and $\dot{\Delta}^\omega$ in (39). We note that the $\omega$-limit of the derivative of the Bethe-Salpeter equation (23) is

$$\dot{\Gamma}^\omega = f \odot \dot{\Delta}^\omega \odot \Gamma^\omega + f \odot R^\omega \odot \dot{\Gamma}^\omega \,,$$

which leads to

$$\dot{\Gamma}^\omega = \Gamma^\omega \, \dot{\Delta}^\omega \, \Gamma^\omega \,,$$

and, substituted into (43), to the compact expression

$$\sigma_{ab}^{\mathrm{H}} = i \, e^2 \, \mathrm{Tr}\Big( J_a^\omega \odot \dot{\Delta}^\omega \odot J_b^\omega \Big) \,, \tag{44}$$

which involves again the $\omega$-limit of the current vertex (41).

Let us finally consider the charge density-density response function. We can repeat step-by-step the above manipulations but now using the $\omega$-limit of the vertex. The final result is that the charge density-density response function reads

$$\chi_{\rho\rho} = \chi_{\rho\rho}^\omega + \mathrm{Tr}\Big( \rho^\omega \odot \big( R - R^\omega \big) \odot \rho^\omega \Big) + \mathrm{Tr}\Big( \rho^\omega \odot \big( R - R^\omega \big) \odot \Gamma \odot \big( R - R^\omega \big) \odot \rho^\omega \Big) \,, \tag{45}$$

where $\rho^\omega$ is the $\omega$-limit of the charge density vertex,

$$\chi_{\rho\rho}^\omega = \mathrm{Tr}\Big( \rho^\omega \odot R^\omega \odot \rho_0 \Big) \,, \tag{46}$$

the $\omega$-limit of the response function, and $\rho_0$ the bare vertex. Charge conservation implies that $\chi_{\rho\rho}^\omega = 0$, while the Ward-Takahashi identities that $\rho^\omega$ is the identity matrix in orbital space [14]. Since $R^q - R^\omega = -\Delta^\omega$, see (39), the charge compressibility is readily obtained as

$$\begin{aligned}
\kappa = -\chi_{\rho\rho}^q &= \mathrm{Tr}\Big( \Delta^\omega \Big) - \mathrm{Tr}\Big( \Delta^\omega \odot \Gamma^q \odot \Delta^\omega \Big) \\
&= -\frac{1}{V} \sum_{\mathbf{k}\ell} \frac{\partial f\big(\epsilon_\ell(\mathbf{k})\big)}{\partial \epsilon_\ell(\mathbf{k})} - \frac{1}{V^2} \sum_{\mathbf{k}\mathbf{k}'\ell\ell'} \frac{\partial f\big(\epsilon_\ell(\mathbf{k})\big)}{\partial \epsilon_\ell(\mathbf{k})} \, \Gamma_{\ell\ell',\ell'\ell}^q(\mathbf{k}, \mathbf{k}') \, \frac{\partial f\big(\epsilon_{\ell'}(\mathbf{k}')\big)}{\partial \epsilon_{\ell'}(\mathbf{k}')} \,,
\end{aligned} \tag{47}$$

which is the conventional result of Fermi liquid theory showing that the compressibility is not just the quasiparticle density-of-states at the chemical potential, but acquires a correction from the quasiparticle interaction.

## B  Quasiparticle Green's function

In this Appendix we show how to rigorously define the quasiparticle Green's function $\hat{G}_*(i\epsilon, \mathbf{k})$ through the fully interacting thermal one $\hat{G}(i\epsilon, \mathbf{k})$ of the physical electrons, having in mind a multiband system in which both Green's functions are matrices. This will give us the opportunity to clarify some results mentioned in the main text.

The physical Green's function $\hat{G}(i\epsilon, \mathbf{k})$ satisfies the Dyson equation

$$\hat{G}(i\epsilon, \mathbf{k})^{-1} = i\epsilon - \hat{H}_0(\mathbf{k}) - \hat{\Sigma}(i\epsilon, \mathbf{k}) \,,$$

where $\hat{H}_0(\mathbf{k})$ is the non-interacting Hamiltonian, not to be confused with the quasiparticle one in (1), and the self-energy $\hat{\Sigma}(i\epsilon, \mathbf{k})$ accounts for all interaction effects. We recall that $\hat{\Sigma}(i\epsilon, \mathbf{k})^\dagger = \hat{\Sigma}(-i\epsilon, \mathbf{k})$, which implies that

$$\hat{\Sigma}_1(i\epsilon, \mathbf{k}) = \frac{1}{2} \Big( \hat{\Sigma}(i\epsilon, \mathbf{k}) + \hat{\Sigma}(i\epsilon, \mathbf{k})^\dagger \Big) = \frac{1}{2} \Big( \hat{\Sigma}(i\epsilon, \mathbf{k}) + \hat{\Sigma}(-i\epsilon, \mathbf{k}) \Big) \,,$$

is hermitean and even in $\epsilon$, while

$$\hat{\Sigma}_2(i\epsilon, \mathbf{k}) = \frac{1}{2i}\left(\hat{\Sigma}(i\epsilon, \mathbf{k}) - \hat{\Sigma}(i\epsilon, \mathbf{k})^\dagger\right) = \frac{1}{2i}\left(\hat{\Sigma}(i\epsilon, \mathbf{k}) - \hat{\Sigma}(-i\epsilon, \mathbf{k})\right),$$

is still hermitean but odd in $\epsilon$. In addition, its eigenvalues are negative for positive $\epsilon$, positive for negative $\epsilon$, and vanish when $\epsilon$ is strictly zero.
One defines [31] a semi-positive definite matrix

$$\hat{Z}(\epsilon, \mathbf{k}) = \left(1 - \frac{\hat{\Sigma}_2(i\epsilon, \mathbf{k})}{\epsilon}\right)^{-1} = \hat{A}(\epsilon, \mathbf{k})^\dagger\,\hat{A}(\epsilon, \mathbf{k}), \tag{48}$$

with eigenvalues $\in [0, 1]$, which plays the role of the quasiparticle residue, and the hermitean frequency-dependent Hamiltonian

$$\hat{H}_*(\epsilon, \mathbf{k}) = \hat{H}_*(-\epsilon, \mathbf{k}) = \hat{A}(\epsilon, \mathbf{k})\left(\hat{H}_0(\mathbf{k}) + \hat{\Sigma}_1(i\epsilon, \mathbf{k})\right)\hat{A}(\epsilon, \mathbf{k})^\dagger, \tag{49}$$

through which

$$\hat{G}(i\epsilon, \mathbf{k}) = \hat{A}(\epsilon, \mathbf{k})^\dagger\,\frac{1}{i\epsilon - \hat{H}_*(\epsilon, \mathbf{k})}\,\hat{A}(\epsilon, \mathbf{k}). \tag{50}$$

The quasiparticle Green's function (3) is simply obtained through the low-frequency limit of $\left(i\epsilon - \hat{H}_*(\epsilon, \mathbf{k})\right)^{-1}$ in (50), which, provided $\hat{H}_*(\epsilon, \mathbf{k}) = \hat{H}_*(-\epsilon, \mathbf{k})$ has a regular Taylor expansion in $\epsilon$, i.e., $\hat{H}_*(\epsilon, \mathbf{k}) \simeq \hat{H}_*(0, \mathbf{k}) + O\left(\epsilon^2\right)$, reads, at leading order,

$$\hat{G}_*(i\epsilon, \mathbf{k}) \simeq \frac{1}{i\epsilon - \hat{H}_*(0, \mathbf{k})} \equiv \frac{1}{i\epsilon - \hat{H}_*(\mathbf{k})}. \tag{51}$$

The *quasiparticle Fermi surface* corresponds to the manifold $\mathbf{k} = \mathbf{k}_{*F}$ in momentum space where

$$\det\left(\hat{H}_*(\mathbf{k}_{*F})\right) = \det\left(\hat{Z}(0, \mathbf{k}_{*F})\right)\det\left(\hat{H}_0(\mathbf{k}_{*F}) + \hat{\Sigma}_1(0, \mathbf{k}_{*F})\right) = 0, \tag{52}$$

which includes the roots of both terms on the right hand side, and where $\hat{Z}(0, \mathbf{k})$ must be evaluated through the limit $\epsilon \to 0$ of (48). If $\det\left(\hat{Z}(0, \mathbf{k})\right)$ is finite we observe that, since

$$G(0, \mathbf{k}) = -\frac{1}{\hat{H}_0(\mathbf{k}) + \hat{\Sigma}_1(0, \mathbf{k})},$$

the roots of $\det\left(\hat{H}_0(\mathbf{k}) + \hat{\Sigma}_1(0, \mathbf{k})\right)$ are actually the poles of $\det\left(\hat{G}(0, \mathbf{k})\right)$ and define the physical Fermi surface, $\mathbf{k} = \mathbf{k}_F$. On the contrary, one realises through (50) that the roots of $\det\left(\hat{Z}(0, \mathbf{k})\right) = \left|\det\left(\hat{A}(0, \mathbf{k})\right)\right|^2$ correspond to those of $\det\left(\hat{G}(0, \mathbf{k})\right)$, which thus define the physical Luttinger surface $\mathbf{k} = \mathbf{k}_L$. More precisely, $\det\left(\hat{\Sigma}_1(0, \mathbf{k})\right)$ has a simple pole on the Luttinger surface, thus $\det\left(\hat{G}(0, \mathbf{k}_L)\right) = 0$, while $\det\left(\hat{Z}(0, \mathbf{k})\right)$ a second order root, so that (52) is indeed verified for $\mathbf{k}_{*F} = \mathbf{k}_L$. It follows that the *quasiparticle Fermi surface* $\mathbf{k}_{*F} = \mathbf{k}_F \cup \mathbf{k}_L$ comprises both physical Fermi and Luttinger surfaces, as we mentioned in the main text.

In reality, (50) is an exact factorisation of the physical electron Green's function that remains valid also in insulators lacking Fermi and Luttinger surfaces, and which was exploited in [10]

to explicitly calculate the winding number $W(G)$ in two dimensions. Indeed, since $W(M\,N) = W(M) + W(N)$ and $W(M) = 0$ if $M = M^\dagger$, then

$$W(G) = W(A^\dagger\,G_*\,A) = W(G_*) + W(A^\dagger\,A) = W(G_*)\,.$$

One can rigorously prove [10] that $W(G_*)$ reduces to the well-known TKNN formula [38] calculated with the eigenstates of $\hat{H}_*(\mathbf{k})$ in (51). The derivation remains simply true even if there is a quasiparticle Fermi surface, a result we used in the Sect. 4

Assuming the case in which a quasiparticle Fermi surface exists, Landau's Fermi liquid theory can be microscopically derived [14, 13] under the sole condition that $\hat{Z}(\epsilon, \mathbf{k})$ and $\hat{H}_*(\epsilon, \mathbf{k})$ are analytic matrix-valued functions of $\epsilon$ in the vicinity of the origin $\epsilon = 0$ and of $\mathbf{k}$ close to the quasiparticle Fermi surface $\mathbf{k}_{*F}$. This condition is verified not only near a physical Fermi surface, $\mathbf{k} \simeq \mathbf{k}_F$, but also near a Luttinger surface [30, 31], $\mathbf{k} \simeq \mathbf{k}_L$, despite the singular self-energy. Moreover, we emphasise that the Nozières and Luttinger derivation [14] is fully non-perturbative, and, therefore, perfectly valid also when a Luttinger surface is present, which does entail a breakdown of perturbation theory.

Since most of the technical details have been already presented in Appendix A, we end by briefly sketching how Nozières and Luttinger derivation [14] works to appreciate its elegance and non-perturbative character. Let us therefore consider the Bethe-Salpeter equations (23) and (29) for the scattering, four-leg, and the current, or, equivalently, the density, two-leg, vertices, now, however, written for the physical electrons. One defines the corresponding quantities for the quasiparticles by contracting each incoming external leg of the physical vertices with $\hat{A}(\epsilon, \mathbf{k})$ and each outgoing one with $\hat{A}(\epsilon, \mathbf{k})^\dagger$. At this stage, these renormalised vertices still depend on the Matsubara frequencies of the external legs.

Having absorbed $\hat{A}(\epsilon, \mathbf{k})$ and $\hat{A}(\epsilon, \mathbf{k})^\dagger$ in the vertices, the kernel $R$ in the Bethe-Salpeter equations (23) and (29) must be replaced by another kernel, denoted as $R_*$, and which, in the basis that diagonalises $\hat{H}_*(\epsilon, \mathbf{k})$ in (49), with eigenvalues $\epsilon_\ell(\epsilon, \mathbf{k})$, has matrix elements, see (50),

$$R_{*\,\ell m}(i\epsilon, \mathbf{k}; i\omega, \mathbf{q}) = \frac{1}{i\epsilon + i\omega - \epsilon_\ell(\epsilon + \omega, \mathbf{k} + \mathbf{q})} \; \frac{1}{i\epsilon - \epsilon_m(\epsilon, \mathbf{k})}\,. \tag{53}$$

One can next proceed similarly as we did in Appendix A, and rewrite all quantities in terms of $\omega$- or $q$-limits of the various vertices. In that way, the kernel $R_*$ is replaced by $R_* - R_*^q$ or $R_* - R_*^\omega$. However, the final expressions of the transport coefficients and thermodynamic susceptibilities only involve $\Delta_*^\omega = R_*^\omega - R_*^q$, which can be finite only because (53) has a non-analytic behaviour at $\omega = q = 0$ due to the quasiparticle Fermi surface, as previously discussed. As a result, one readily finds [14] that, regarded in the sense of a distribution,

$$\Delta_{*\,\ell m}^\omega(\epsilon, \mathbf{k}) \equiv \frac{\delta_{\epsilon,0}}{T}\,\Delta_{\ell m}^\omega(\mathbf{k})\,, \quad \dot{\Delta}_{*\,\ell m}^\omega(\epsilon, \mathbf{k}) \equiv \frac{\delta_{\epsilon,0}}{T}\,\dot{\Delta}_{\ell m}^\omega(\mathbf{k})\,, \tag{54}$$

where $\Delta^\omega$ and $\dot{\Delta}^\omega$ are just the quasiparticle ones in (39), and all vertices must correspondingly be evaluated at zero Matsubara frequency, thus reducing to those in Appendix A. This is in essence the microscopic proof of the equivalence between susceptibilities and transport coefficients of the physical electrons and those of the quasiparticles treated within HF+RPA.

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
