# Peer review of "Fermi-liquid corrections to the intrinsic anomalous Hall conductivity of topological metals"

_SciPost Physics_

## Round 1 · Referee Report · Anonymous (Referee 1) · 2024-9-28

Report
The authors pose and analyze a very interesting and relevant question about Fermi-liquid corrections to the intrinsic anomalous Hall effect in metals. About 20 years ago, Haldane has shown that in the noninterating Fermi gas, a fractional part of the Hall conductivity, given by the integral of the Berry curvature over the Fermi volume, is in fact a Fermi surface property as it reduces to a Fermi surface integral of the Berry connection. It is therefore tempting to assume that the same holds true in the interacting Fermi liquid because at the Fermi surface the quasiparticles and their wave functions, determining the Berry connection, remain well-defined. The authors challenge this seemingly natural assumption and argue that the residual interaction between quasiparticles generate nontrivial (vertex) corrections to the Haldane formula for the Hall conductivity. This result may have serious implications not only for topological materials near the Mott transition, as the authors discuss, but also for the first principles calculations of the Hall conductivity in usual ferromagnets. As the Landau parameters in metals are typically not small, the standard Berry curvature based results for the intrinsic contribution to $\sigma_H$ become somewhat questionable and should be reconsidered. Therefore, the presented results, if correct and presented a bit more clearly (see below), can justify publication in SciPost in accordance with the last two items in the list of expectations (items 3 and 4 as indicated by the authors).
I personally find the main statement of this work very plausible physically. I like very much a simple analogy to the Drude weight in metals. When computing linear transport coefficients, the external current vertices should be taken in the "nonequilibrium" $\omega$-limit, which makes the appearance of interaction (vertex) corrections unavoidable. Despite I tend to agree with the final result, I think there are several problematic points in the presentation and argumentation, which should be clarified before a possible acceptance.
I. General questions to the abstract and introduction
-
In the abstract, the authors state that the corrections they discuss are already accounted for by the "Landau's Fermi liquid theory of topological metals". However, it is not explained in the text what this actually mean. Moreover, it seems this term does not even appear in the main text.
-
In the introduction, the authors quote, but do not discuss, the paper of Ref.[11] by Chen and Son. However, it looks very relevant to the present work. As far I can see, in Ref.[11] Fermi-liquid interaction corrections to the Berry phase expression for the Hall conductivity are discussed, calculated, and attributed physically to a "dipole moment of the quasiparticles". It is not clear to me whether it is exactly the same or not, but I think in such a situation a more detailed discussion of the relation to Ref.[11] is necessary.
II. Technical points which require clarification
-
I think the basic statement of the problem in Sec.2 should be explained better. What mean $\alpha, \beta$ indexes in eq.(1). I understand that the authors probably have in mind tight-binding or similar models, but what it could be in the real world. Should we understand $H_*(k)$ as a differential operator in real space, i.e. the Hamiltonian of a crystal in the k-representation defined on a unit cell with periodic boundary conditions. If so, it has an infinite number of discrete eigenvalues for each k in the BZ -- the crystal's band structure. It makes a sense to indicate this more clearly, as considering infinitely many discrete eigenvalues (the bands) at each k looks highly unusual in the context of the Fermi liquid theory.
-
If the above understanding is correct, then it is probably also correct that the physical significance have only the eigenstates of $H_*(k)$ which belong to the Fermi surface, while the overwhelming majority of states/bands that appear in the present formalism, but have energies away from the Fermi energy, are strictly speaking unphysical. In my opinion, this also requires a clarification, especially in connection to the final formulas (8)-(9) for the Hall conductivity, which explicitly involve integration over the unphysical states. What looks to me quite disturbing is that these states appear as eigenstates of a fictitious eigenvalue problem with Hamiltonian $H_*(k)$. Possibly these fictitious states somehow disappear at the very end, as it apparently happens in the specific toy example in Sec.3. I believe this issue is serious enough to be discussed in some length, especially in the situation when the general demonstration of this fact is absent.
-
It is in general a very good idea to present more technical details and rigorous arguments/proofs in appendices. In particular, Appendix B is aimed at justifying the approach used in Sec.2 and detailed in Appendix A. In my opinion, the most delicate point in the whole paper is the calculation of the linear response (current-current correlation) function using the "pseudo-HF" Green function (3) with a frequency independent self-energy. Unfortunately, precisely this point remains unexplained. The justification relies on eq.(14), which is indeed obvious for n=l at the Fermi surface, but does not look trivial if n is not equal to l. For the off-diagonal terms the electron-hole excitations are gapped, which makes the corresponding contribution non-singular and therefore the arguments typically used in similar derivations, for example in Ref.[14], do not apply, at least straightforwardly. If eq.(14) and a formal justification of the applied methodology can indeed be "readily found", it should be presented. I believe this will allay most of my concerns.
After the requested clarifications and explanations, I would probably recommend this work for the publication. In general, the manuscript is sufficiently well written, cites the relevant literature, and satisfies (up to the above listed points) the standard criteria required for a good scientific work.
Recommendation
Ask for major revision

---

## Round 1 · Referee Report · Anonymous (Referee 2) · 2024-11-12

Strengths
- Non-perturbative results for Hall conductance in quantum many-body problem
- Interesting new strategy to access ongoing questions in strongly interacting topological insulators (by considering them as un-doped Fermi liquids)
- Paper well structured in firm results (Secs. 2&3) and interesting conjectures (conclusions)
Weaknesses
- Technicalities are a bit cumbersome (I would have found a case study for a continuum system with full rotational symmetry more intuitive than the lattice model)
Report
The paper on Fermi liquid corrections by Pasqua and Fabrizio addresses a very topical and at the same time long-standing question of strongly interacting anomalous Hall metals by carefully taking into account corrections due to residual Fermi liquid interactions.
If correct, this work provides a substantial step forward beyond previous results and may allow to connect the Berry Fermi liquid theory with the insulating counterpart. As such, I would agree with the authors that the manuscript ticks the boxes of groundbreaking theoretical discovery/theoretical breakthrough.
However, I have a multitude of remarks/questions to be answered before I can take a final decision.
In the answer to the referees, could the authors please elaborate on the following question?
- Can the authors comment on the insufficiency of previous works, in particular of the kinetic equation based approach by Chen and Son, Ref [9]? What is missing there?
(For requested changes, please see below)
Requested changes
-
Regarding the discussion part: It is well know that the discrepancy between Chern and winding number can occur in the presence of fractionalization, e.g. quantum spin liquids. These can also be be understood by sending certain Fermi liquid parameters to -2 while keeping other FL parameters finite, cf RMP 89, 025003 (2017) and references therein. I feel this analogy should be discussed more clearly in the paper.
-
I find the statement that the "intrinsic anomalous Hall conductivity ... is... genuinely a property of the quasiparticle Fermi surface" very dangerous. Already the non-interacting, clean Berry curvature contribution can be converted from Fermi surface to Fermi sea contribution by a single partial integral. In fact, the Berry curvature is really a manifestation of interband coherence (see Eq. (9)).
2a) Above all, I ask the authors to explain to the readers why Fermi liquid theory (which is geared to only make statements about the Fermi surface and its vicinity) can be trusted in this calculation of interband effects.
2b) Beyond that, I propose to adapt the wording in the introduction.
- The referencing of works on topological Green's function zeros is rather incomplete. In particular, I find that the following articles from various different groups and using a variety of different techniques are important for this work but missing: a. PRB 90, 060502(R) (2014), b. PRL 133, 126504 (2024), c. PRL 133, 136504 (2024), d. arXiv:2405.08093.
Recommendation
Ask for minor revision

---

## Round 2 · Referee Report · Anonymous (Referee 1) · 2025-3-3

Report

I am happy to see that the authors very seriously considered all points raised in the referee reports. In the revised manuscript, all my questions and concerns have been addressed absolutely satisfactory. Therefore, I can now recommend publication of this work in SciPost.

I should admit that the appearance of some formal deep bands/states in the final formulas still look to me quite counterintuitive. However now the authors added a detailed appendix supporting their multiband generalization of the Fermi liquid theory. If indeed correct, this is very interesting, though-provoking, and goes far beyond the specific problem of interaction/Fermi-liquid corrections to the Hall conductivity.

Recommendation

Publish (easily meets expectations and criteria for this Journal; among top 50%)

---

## Round 2 · Author Response

Dear Editor,
Thanks for sending us the two referee reports, which we carefully read and whose valuable comments and suggestions are incorporated into the revised version we are submitting here. It took us some time to extend the appendices to meet the Referees’ requests, particularly Appendix C.2, which is a crucial result in the multi-band generalisation of Landau’s Fermi liquid theory in the presence of a non-trivial topology. While it may appear lengthy and cumbersome, it is actually a straightforward application of perturbation theory for non-interacting electrons. We hope that the revised version will meet the expectations of both referees.

Yours sincerely,
Ivan Pasqua and Michele Fabrizio

Reply to the first Referee

First, we would like to express our sincere gratitude to the Referee for taking the time to thoroughly review our manuscript and providing us with such precise suggestions that have significantly enhanced the clarity of our work.
In the following, we will address each point raised in the report in detail.

I.1) Following Referee’s suggestion, we replace “already” with “naturally” and erase “of topological metals” from the abstract. Specifically, we now write “Furthermore, it demonstrates that such corrections are naturally accounted for by Landau's Fermi liquid theory, here extended to the case in which coherence effects between bands crossing the chemical potential and those that are instead away from it may play a crucial role, as in the anomalous Hall conductivity, …”, since, following a request by both Referees, we have substantially extended the Appendix, in particular adding Appendix C where we present a detailed microscopic derivation of Fermi Liquid theory in the realistic case of metals with many bands, only some of which cross the chemical potential, which is necessary to discuss topology. This is also the reason why it took us some time to resubmit the manuscript.

I.2) In the revised manuscript, we have delved deeper into the crucial work by Chen and Son, both in the Introduction and at the conclusion of Section 3. In their research, Chen and Son derive a linearised kinetic equation for a Fermi liquid with a Berry curvature. They identify corrections to the anomalous Hall conductance that they attribute to an electric dipole moment carried by the quasiparticles. On the contrary, the Fermi liquid corrections we derive within our formalism, as highlighted by the Referee, merely reflect the distinction between the static and dynamic limits of the linear response functions. While we could not find a straightforward physical argument to support this, we think it is possible that our findings align with the interpretation proposed by Chen and Son.

II.1) We agree with the Referee that the idea of discussing Fermi liquid theory from an ab-initio Hamiltonian that includes an infinite number of bands initially appears somewhat peculiar, even though the primary focus is on bands crossing the chemical potential and those responsible for the non-trivial topology. Nevertheless, the formalism presented in Appendix B is highly general and serves as the foundation for deriving Landau’s Fermi Liquid theory in Appendix C. Surprisingly, we found that this derivation appears to be valid regardless of the number of bands, particularly in the context of the anomalous Hall conductivity discussed in Appendix C.2. In fact, we thoroughly checked the calculations in that Appendix, which are lengthy and tedious, but ultimately consist solely of manipulating perturbation theory for non-interacting electrons described by a Hamiltonian where, see Appendix B, the Matsubara frequency just plays the role of an additional Hamiltonian parameter. Such calculations are commonly performed in all works that discuss the topological properties of solids, assuming the absence of electron-electron interaction, in which case, too, the formalism remains general and independent of the number of bands.

II.2) We revised the statement at the end of Sec. B, as well as the beginning of the Conclusions section, to clarify the point raised by the Referee. We believe, in fact, that equations (8), (9) and (10) have a more general validity, consistently with what we write above in point II.1). Indeed, it might well happen that an occupied band has still a topological character and thus contributes to Eq. (8) but with a non-quantised value because of (10), i.e., because of the interaction with the quasiparticles at the Fermi surface. This result is not wrong, as one might believe, since the contribution of the occupied band becomes quantised again if no band crosses anymore the chemical potential. Put it differently, we believe that our results, equations (8) to (10), constitute a generalisation of Haldane’s results, which, e.g., would imply that if the band crossing the chemical potential is not topological while an occupied band is, yet the anomalous Hall conductivity is not necessarily quantised.

II.3) As discussed earlier, in the revised manuscript, we have included Appendix C, which presents a formal and detailed microscopic derivation of Landau’s Fermi liquid theory for multiple bands. This derivation is a novel result that we believe is significant and surpasses the standard single-band case. We want to emphasise that, apart from the standard Fermi liquid assumptions, we make only one additional assumption: that a scattering process between distinct bands involving the dynamic correction to the current does not preserve phase coherence and averages to zero when summed over frequency, momentum, and band indices. This point is extensively discussed in Appendix C.2. Briefly, an interband transition induced by the dynamic correction to the current is a scattering process due to interaction between a quasiparticle-quasihole pair on two distinct bands at finite , and an intraband pair at on the Fermi surface. The phase coherence between the two pairs at different frequencies is not guaranteed, rather the contrary. Therefore, we argue that, whenever one has to sum over frequency, momentum and bands, this interaction-induced interband transition yields full decoherence and thus averages at zero. On the contrary, in all quantities that catch just the singularities at that arise from the discontinuity of the phase of the Green’s function, see Appendix C.2 for the case of the anomalous Hall conductivity, the above pairs are both at zero frequency and phase coherence is maintained.

We once again express our deepest gratitude to the Referee for the valuable feedback. We sincerely hope that the revisions address all concerns satisfactorily.

Reply to the second Referee

First, we would like to express our sincere gratitude to the Referee for taking the time to carefully review our manuscript and providing us with such precise suggestions that have greatly aided us in enhancing the presentation of our results.
In the following, we will address each point raised in the report in detail.

1.) In the revised manuscript, we failed to properly cite the work by Chen and Son. We have rectified this error and now give full credit to their work, as evident in the revised Introduction and the conclusion of Section 3. Nevertheless, despite the possibility that the corrections Chen and Son identify, which they attribute to a quasiparticle electric dipole, and our own, which stem from the distinction between static and dynamic limits in a metal, are related, we were unable to find a physical argument to support this claim.

Concerning the requested changes:

1.) We agree with the Referee, and apologise for not having emphasised the natural connection with gapless quantum spin liquids. We now mention the relationship in the Conclusions section and cite the relevant review pointed out by the Referee.

2.) We perfectly agree with the Referee and modified accordingly that sentence.

2.a) Since the same question was raised by the first Referee, the revised version includes a new Appendix C. In this appendix, we present a detailed microscopic derivation of Landau’s Fermi liquid theory. This theory generalises the one by Nozières and Luttinger when the interaction between bands that cross the chemical potential and those that are either fully occupied or empty plays a crucial role. This is precisely the case of the anomalous Hall conductivity. This is also the reason why it took us time to resubmit the manuscript. Since this derivation is noteworthy on its own, we refer to it more extensively in the text. In fact, we already mention it in the abstract, where we borrow an enlightening sentence from the Referee, with their permission. We emphasise that the only additional assumption that we make besides the standard Fermi liquid ones is that a scattering process between distinct bands that involves the dynamic correction to the current does not maintain phase coherence, and thus averages at zero upon summing over frequency, momentum and band indices. This point is discussed at length in Appendix C.2. Briefly, an interband transition induced by the dynamic correction to the current is a scattering process due to interaction between a quasiparticle-quasihole pair on two distinct bands at finite , and an intraband pair at on the Fermi surface. The phase coherence between the two pairs at different frequencies is not guaranteed, rather the contrary. Therefore, we argue that, whenever one has to sum over frequency, momentum and bands, this interaction-induced interband transition yields full decoherence and thus averages at zero. On the contrary, in all quantities that catch just the singularities at that arise from the discontinuity of the phase of the Green’s function, see Appendix C.2 for the case of the anomalous Hall conductivity, the above pairs are both at zero frequency and phase coherence is maintained.

3.) We thank the Referee for pointing to our attention relevant works that we originally missed, and which we cite in the revised version.

We once again express our gratitude to the Referee for providing valuable feedback. We sincerely hope that the revisions address all concerns satisfactorily.

---

## Round 2 · List of Changes

1) We changed the abstract following very useful suggestions from the Referees;
2) We have substantially extended the Appendix, in particular adding Appendix C where we present a detailed microscopic derivation of Fermi Liquid theory in the realistic case of metals with many bands, only some of which cross the chemical potential, which is necessary to discuss topology;
3) We have delved deeper into the crucial work by Chen and Son, both in the Introduction and at the conclusion of Section 3;
4) We now mention the natural connection with gapless quantum spin liquids in the Conclusions section and cite the relevant review;
5) We cite relevant works that were missing in the first version of our work.

---

## Editorial Decision

accepted_in_target_journal